# Is Osteoarthritis Always Associated with Low Bone Mineral Density in Elderly Patients?

**DOI:** 10.3390/medicina58091207

**Published:** 2022-09-02

**Authors:** Bojana N. Stamenkovic, Natasa K. Rancic, Mila R. Bojanovic, Sonja K. Stojanovic, Valentina G. Zivkovic, Dragan B. Djordjevic, Aleksandra M. Stankovic

**Affiliations:** 1Institute for Treatment and Rehabilitation Niska Banja, Rheumatology Clinic, 18000 Nis, Serbia; 2Faculty of Medicine, University of Nis, 18000 Nis, Serbia; 3Institute for Public Health Nis, 18000 Nis, Serbia; 4ENT Clinic, University Clinical Center of Nis, 18000 Nis, Serbia

**Keywords:** osteoarthritis, bone mineral density, osteoporosis

## Abstract

*Background and Objectives*: The relationship between osteoarthritis (OA) and osteoporosis (OP) has been analysed for over four decades. However, this relationship has remained controversial. Numerous observational and longitudinal studies have shown an inverse association between the two diseases and a protective effect of one against the other. On the other hand, some studies show that patients with OA have impaired bone strength and are more prone to fractures. The study’s main objective was to determine the bone mineral density (BMD) of the spine and hip (femoral neck) of postmenopausal women of different ages, with radiologically determined OA of the hip and knee, as well as to determine the correlation between BMD values and age in the experimental group. *Materials and Methods*: The retrospective cohort study included 7018 patients with osteoarthritis of peripheral joints and the spine, examined by a rheumatologist in an outpatient rheumatology clinic at the Institute for Treatment and Rehabilitation, Niška Banja from July 2019 to March 2021. A nested anamnestic study was conducted within the cohort study of patients, and it included two groups: an experimental group composed of 60 postmenopausal women, and a control group composed of the same number of women. Out of 120 patients, 24 did not meet the criteria for the continuation of the study (due to technical errors—radiographic and/or densitometry artefacts). Fifty-six postmenopausal women (aged 45–77 years) with hip and knee radiological OA were examined as an experimental group. The participants were divided into two subgroups according to age (45–60 years and over 61 years). The control group included 40 healthy postmenopausal women of the same age range, without radiological OA, with normal BMD of the hip and spine. All patients with OA met the American College of Radiology (ACR) criteria. OA of the hip and knee was determined radiologically according to Kellgren and Lawrence (K&L) classification, and patients were included in the study if a K&L grade of at least ≥ 2 was present. Hip and spine BMD was measured by dual-energy X-ray absorptiometry (DXA). *Results:* Compared to the control group, we found statistically significantly lower BMD and T-scores of the spine in older postmenopausal women: BMD (g/cm^2^), *p* = 0.014; T-score, *p* = 0.007, as well as of the hip: BMD (g/cm^2^), *p* = 0.024; T-score *p* < 0.001. The values of BMD and T-score of the spine and hip are lower in more severe forms of OA (X-ray stage 3 and 4, according to K&L), *p* < 0.001. We found negative correlation between BMD and T-score and age only for the hip: BMD (g/cm^2^), *ρ* = 0.378, *p* = 0.005; T-score *ρ* = −0.349, *p* = 0.010. *Conclusions*: Older postmenopausal women with radiographic hip and knee OA had significantly lower BMD of the hip and spine as compared to the control group without OA, pointing to the need for the prevention and treatment of OA, as well as early diagnosis, monitoring, and treatment of low bone mineral density.

## 1. Introduction

Osteoarthrosis (OA) and osteoporosis (OP) are very common diseases that present a significant socioeconomic problem nowadays due to reduced quality of life of patients and high rates of disability. These conditions are very common in the population over 65 years of age. About 30% of patients over 65 years of age have radiographic evidence of OA [1]. Globally, OA is the sixth leading disabling condition, which accounts for nearly 3% of the total global years of living with disability [2]. A systematic analysis of the Global Burden of Disease Study (2020) pointed to a high prevalence and incidence of OA in 195 countries worldwide (prevalence 3754.2, yearly incidence 181.2 per 100,000 people) with a recorded 9.2% increase in prevalence and an 8.3% increase in incidence since 1990 [3]. Age and female gender are significantly correlated with the disease [4,5]. OA and fragility fracture due to OP represent a significant health burden in the elderly population. The frequency and mortality of the disease can be reduced by removing the risk factors, primarily by reducing body weight and preventing fractures and injuries.

Osteoporosis is a metabolic disease characterised by reduced bone mineral density, disorders of the microarchitecture of bones, and increased susceptibility to fractures. It also represents a significant health and socioeconomic problem, predominantly for the elderly. Lower bone density and osteoporosis are common in postmenopausal women due to estrogen deficiency, but these conditions are also common in older men. Worldwide, 200 million people suffer from this disease, the most severe consequence of which is fractures. Global disability and mortality from this disease have increased from 207,367 and 8,588,936 cases in 1990 to 437,884 and 16,647,466 in 2019 (111.16% and 93.82%, respectively). One in three women and one in five men over 50 years of age will experience a fracture during their lifetime, and each fracture significantly increases the likelihood of a new occurrence or multiple fractures. The residual lifetime risk of fragility fracture is 44% in women and 30% in men aged 60 years or older [6,7]. 

It is estimated that by 2050, the number of spinal fractures will increase by 310%, and the number of hip fractures will increase by 240%. This is explained to a certain extent by the world population’s extended lifespan and general ageing. Compared to the data from 1990, the 2019 data show a significantly increased percentage of fractures caused by the disease in India, China, USA, Japan, and Germany (25.59%, 18.75%, 8.35%, 3.29%, and 3.04%, respectively) [7,8]. Hip fracture is the most dramatic consequence of OP. Literature data show that 10–20% of OP patients do not survive one year following hip fracture, 50% remain permanently disabled, and only 30% manage to recover completely.

Countries with the highest standardised yearly rates of hip fracture incidence in women include Denmark, Sweden, and Austria, while the lowest rates were recorded in Morocco, Ecuador, and Tanzania [9]. Although the prevalence of OA and OP is high, their association is not yet clear and understood [10]. 

The relationship between OA and OP has been analysed for over four decades. Numerous observational and longitudinal studies have shown an inverse association between the two diseases and a protective effect of one against the other. This has been supported by the fact that different mechanisms lead to bone changes in OA and OP. In OP, there is a decrease in bone mass and bone strength, which is associated with fractures, whereas in OA, bone density is increased, and cartilage remodelling takes place. On the other hand, some studies show that patients with OA have impaired bone strength and are more prone to fractures [11,12,13]. 

OA is a pathological process in which changes occur in all joint structures, not only in the cartilage, as considered until recently. The subchondral bone is also the location where various changes occur during the development phases of the disease. In the early phases of OA, osteoclast activity is increased and later, with compact bone thickening and reduced bone mineralisation and elasticity, BMD increases without the accompanying improvement in bone quality [14]. 

Genetic, systemic, and local factors explain the inverse relationship between OA and OP. Concerning the genetic factors, current research deals with the WNT signalling pathways with different genetic variations (LRP5, WNT16, and SOST), genetic variation for cathepsin K, and the analysis of Mendelian randomisation. The systemic effects are realised through hormonal factors and bone metabolism markers (leptin, osteoprotegerin), whereas the local effects are related to the influence of various cytokines and inflammation markers. 

Even though OA can be defined based on symptoms, clinical examination, or radiological changes, or using a combination of the three methods, the studies which have analysed the relationship between bone mineral density (BMD) and OA almost exclusively used the Kellgren and Lawrence (K&L) grading scale to determine radiological changes in the knee or hip [15]. Radiological OA of the knee and hip with a finding of osteophyte/s was connected with increased BMD of the lumbar spine, femoral neck, and/or entire body [16,17,18,19,20,21,22]. However, when radiological OA was defined only by the presence of joint space narrowing, most research did not determine the connection with BMD increase [19,20]. Only one longitudinal study has proved the association between increased BMD and joint space narrowing [20]. However, it seems that the increase in bone mineral density with radiological changes does not impact the reduction of fracture risk [21] and can be associated with increased fracture risk [22]. Some data show that women with knee and hip radiological OA have less bone loss in the lumbar spine and femoral neck compared to women without OA. Individuals with radiological OA (hip and knee for women, hip for men) showed more significant bone loss in the femoral neck during a two-year follow-up [16]. These studies used K&L grading of radiological changes in OA without assessing joint space narrowing and osteophytes separately [20,22,23,24]. The above-stated facts and contradictory findings in lower limbs make us reconsider whether OA in the elderly is always associated with low bone density. Radiological changes in the hip (particularly in the femoral neck) have less influence on BMD than radiological changes in the spine; thus, hip BMD is optimal for predicting osteoporotic fractures [25]. Unfortunately, few studies have analysed hip BMD in terms of interpreting a connection between OA and OP. 

Although earlier studies have confirmed that OA of the upper limbs is not connected with reduced BMD, the latest studies have shown that radiological OA of the hands is connected with reduced BMD and hand, radius [26,27,28,29], and spine [30] bone mass. Intervertebral space narrowing is associated with higher fracture risk [30]. 

Study results vary considerably among regions, the method and technique of measuring bone mass and the types of joints affected by OA (different speed of weight-bearing joint regeneration compared to the speed of regeneration of other joints). 

The relationship between OA and OP also varies depending on whether a patient has a primary generalised form of OA or a localised form [31]. 

There are still different opinions and contradictory results in the relevant literature. Since no similar research has been conducted in Serbia, we decided to explore the problem and determine whether knee and hip OA is associated with low BMD in older postmenopausal women. 

The study’s main objective was to determine the BMD of the spine and hip (femoral neck) of postmenopausal women of different ages, with radiologically determined OA of the hip and knee, as well as to determine the correlation between BMD values and age in the experimental group. 

## 2. Material and Methods

### 2.1. Study Design

We conducted a retrospective cohort study which included all patients examined by a rheumatologist in the outpatient rheumatology clinic at the Institute for Treatment and Rehabilitation, Niška Banja from July 2019 to March 2021. A total of 21,740 patients were examined, out of whom 7018 had osteoarthritis of peripheral joints and spine. In order to create a nested anamnestic study, the inclusion and exclusion criteria were determined for further analysis. The inclusion criteria were:-Female sex, age 44 to 77-Postmenopausal status (no menstrual cycle for 12 months)-Presence of clinical, radiological, and laboratory OA parameters-Participants from southeast Serbia-Consent to participate in the research

The exclusion criteria were the following:-Presence of comorbidities (metabolic diseases, hyperthyroidism, diabetes mellitus bone metastases, rheumatoid arthritis, kidney and liver insufficiency, malabsorption)-Body mass index <19 kg/m or >30 kg/m-Premenopausal and perimenopausal status-Consent for participation was not obtained

A nested anamnestic study was conducted within the cohort study, and it included two groups: an experimental group composed of 60 postmenopausal women, and a control group composed of the same number of women. 

Out of 120 patients, 24 did not meet the criteria for the continuation of the study (due to technical errors—radiographic and/or densitometry artefacts). Fifty-six postmenopausal women aged 45–77 years were included in the study as the experimental group. The experimental group participants were classified as having radiological OA of the hip and knee, according to American College of Radiology (ACR) criteria [32]. They were divided into two subgroups according to age. The first group included patients aged 45–60 years, and the second group included patients older than 61. The control group consisted of 40 postmenopausal women of the same age, with the same inclusion (without OA) and exclusion criteria as the experimental group from the same southeast region of Serbia. They were divided into two age subgroups, similarly to women in the experimental OA group. The control group examinees did not have radiological OA of the hip and knees and had normal values of BMD in g/cm^2^ and T-score of the spine and hip. 

The local medical ethics committee of the Institute for Treatment and Rehabilitation Niška Banja, Serbia approved the study (number of decision 6365/1, dated 17 July 2010). 

### 2.2. Measurements

Clinical examination, radiography of the hip and knee, and DXA densitometry were carried out on all the examined and control group patients from July 2019 to March 2021 at the Clinic for Rheumatology of the Institute for Treatment and Rehabilitation, Niška Banja. We determined the body mass index (BMI) of all examinees. The radiological examination included anteroposterior radiography of the pelvis and both knees in a semiflexed and a standing position. A radiologist interpreted the results without insight into clinical diagnosis and BMD using the K&L grading scale for OA without assessing joint space narrowing and osteophytes separately. OA of the hip and knee was determined by radiological examination according to Kellgren and Lawrence (K&L), and patients were included in the study if at least a grade ≥ 2 K&L was present. 

BMD was measured by dual-energy X-ray absorptiometry (DXA) for the spine and hip (femoral neck) on a Hologic Discovery densitometer. Bone density was expressed as an absolute value in g/cm^2^ and as T-score (aberration in standard deviations compared to a young, healthy population). Diagnosis of OP was made using the World Health Organization T-score criteria using a Caucasian reference population. OP was defined as a T-score lower than −2.5 for either the femoral neck (FN) or lumbar spine (LS). Osteopenia was defined as a T-score between −2.5 and −1.0. The coefficient of variation percentage (CV%) of the lumbar spine (L1–L4) and femoral neck were 1.1% and 0.6%. Every morning additional quality controls were done with the DXA equipment according to the manufacturer’s guidelines to verify the system’s stability, which did not show any shift during the entire study period. The measurements were performed in all groups, with participants on an examination table in the supine position with their limbs abducted away from their trunk [33]. 

### 2.3. Statistical Analysis

All statistical analysis of data was done using SPSS 23.0 for Windows. All continuous variables are given as means ± standard deviations (SD). The average values of continuous variables in the two tested groups were compared by the Student’s *t*-test or Man-Whitney test (depending on data distribution normality for continuous variables). Assessment error less than 5% (*p* < 0.05) was accepted as a statistically significant threshold. Spearman’s correlation coefficient (*ρ*) measures the strength and direction of association between two ranked variables.

## 3. Results

Baseline characteristics and radiological changes of the experimental and control groups of patients are shown in Table 1. There were no statistical differences in age, height, weight, BMI, or smoking status between the examined groups. There were significant statistical differences in radiological changes between groups.

The mean value of spine and hip BMD in g/cm^2^ in both the experimental OA and control group is shown in Table 2. The difference in mean values of BMD in g/cm^2^ in a younger subgroup (aged 45–60 years) between the experimental and control group was not statistically significant for the spine and hip, while the spine and hip BMD values in women older than 61 years of age were statistically lower in experimental OA patients as compared to controls. Lumbar spine BMD (61 years of age and older): 0.942 ± 0.132 vs. 1.068 ± 0.131 g/cm^2^; *p* = 0.014; hip BMD (61 years of age and older): 0.790 ± 0.144 vs. 0.899 ± 0.134; *p* = 0.024. T-score of the spine was significantly lower in the younger OA experimental subgroup compared to controls: lumbar spine T-score (50–61 years of age): −1.110 ± 1.216 vs. −0.483 ± 0.703; *p* = 0.002 (Table 2 and Figure 1). As for the experimental OA group, T-score was significantly lower for the spine and hip in the older subgroup: lumbar spine T-score (61 years of age and older): −1.232 ± 1.070 vs. −0.227 ± 1.006, *p*|= 0.007; hip T-score (61 years of age and older): −1.36 ± 1.04 vs. 0.08 ± 0.75, *p* < 0.001 (Table 2 and Figure 1).

Figure 1 shows the 95% confidence intervals (95% CI) for means in Table 2. It can be seen that when the difference between the groups is significant, either the overlap of the confidence intervals is small or there is no overlap. For example, in the group of women aged 61 years and older, when the BMD scores for the hip are compared, there is a significant difference between the groups (*p* = 0.003) and the overlapping of confidence intervals is very small. In other instances, when the difference between compared groups is not significant, the overlapping of the confidence intervals is larger. For example, in the group of women aged from 45 to 60 years, when the BMD scores for the spine are compared, there is no significant difference between the groups (*p* = 0.416) and the overlapping of the confidence intervals is large. This observation was important for deriving final conclusions. 

The values of BMD and T-scores of the spine and hip are lower in more severe forms of OA (X-ray stage 3 and 4, according to K& L, *p* < 0.001 (Table 3).

Negative correlation between age and values of BMD (g/cm^2^), *ρ* = −0.378, *p* = 0.005 and T-score, *ρ* = −0.349, *p* = 0.010 was determined only for the hip (Table 4 and Figure 2).

The radiological findings of the knee in PA projection and the DXA findings of the spine and hip in the same patient with OA and OP are presented in Figure 3. Osteoarthritis of a significant degree is shown on the right knee: asymmetric narrowing of the joint fissure, beaked circular osteophytes, and spinous processes on the intercondylar eminence. A poorly defined zone of demineralisation is shown under the lateral tibial plateau. A similar morphology finding is significantly more discrete for the left knee.

## 4. Discussion

This research examined the value of BMD in postmenopausal women from different age subgroups with radiological OA of the hip and knee and showed a significantly reduced BMD of the spine and hip in an older subgroup of patients (older than 61 years of age). This is in line with earlier clinical observations and epidemiology studies [17,18,19] which showed an inverse connection between OA and osteoporosis. However, most of these studies measured the spine and/or proximal femur BMD. The presence of osteophytes, vertebral endplate sclerosis, and vascular calcification can increase spine BMD, thus masking bone loss which has developed due to ageing or disease. Our results confirmed that observation, as we recorded a negative correlation between BMD and age only for the hip, not the spine. Even though radiological OA of the knee and hip are in correlation with radiological OA of the lumbar spine [34], it is logical that a positive association between knee or hip osteophytes and lumbar spine and BMD is a consequence of local degenerative changes. 

Earlier studies have shown that the femoral neck is a valid location for BMD evaluation, and in line with those findings we monitored BMD of the spine and hip (femoral neck) in this study. 

Our results can be compared to other controversial results recorded in the literature. Increased BMD of the lumbar spine, femoral neck, and/or entire body has been determined by radiological OA of the knee and hip with osteophytes [16,17,18,19,20,35,36]. However, radiological OA with joint space narrowing did not correlate with BMD increase [10,11]. Limited mobility and pain in severe OA can accelerate bone loss and lead to osteoporosis. Our results showed that bone mineral density was lower in severe hip and knee OA. 

More recent studies have shown that radiological OA of the hands is associated with decreased BMD and bone mass of the hands, radius [26,27,28,29], and spine [30]. 

It was determined that the risk of vertebral fracture was not reduced in postmenopausal women with spine OA, even though the patients had high BMD values. It was determined that the narrowing of the intervertebral (IV) space contributed to increasing risk of vertebral fractures [37] in these women. However, spine BMD values cannot be a relevant surrogate marker for assessing OP of the spine in patients with OA [38]. Hip OA, defined by joint space narrowing, significantly correlates with BMD reduction of the hip, while osteophytes of the hip impact the increase in BMD, which is not statistically significant. Multiannual monitoring of individuals with OA and joint space narrowing of the hips and knees has determined a relation between the narrowing and more significant bone loss. The presence of osteophytes on the hips and knees, which has been associated with the non-significant increase in BMD at the beginning of the study, is associated with more significant bone loss in the hip [39]. In the last ten years, several studies have indicated an association between increased BMD and early changes in knee and hip cartilage, which was explained by genetic factors. In 2017, results were published of the prospective monitoring of BMD and changes in cartilage for two years in healthy middle-aged individuals without clinically evident OA using MR. It was shown that deterioration and cartilage volume loss are faster in people with increased BMD [40]. 

The results of this study point out that radiological OA is connected with bone loss, which indicates the possibility of applying antiresorptive therapy to OA evaluation [39,40]. After a three-year follow-up, strontium ranelate has shown positive effects in preventing joint space reduction in patients with OA, as it has had effects on the bone, not on joint cartilage [41]. Insufficient evidence of cardiovascular safety during the medication application has resulted in caution regarding the further application of this medication [42]. The expected potential positive effect of bisphosphonate and cathepsin K in other studies has not been achieved, with an explanation that there is a need for patient phenotyping according to the degree of radiological bone changes and osteophytosis for the purpose of adequate treatment of OA [43]. 

The mechanism of association of OA and OP has not yet been established. The hypothesis that uses the phenotype of a person to explain changes in early and advanced OA and determine the association with osteoporosis is 15 years old. A hypertrophic, osteoformative type with increased BMD and slow OA progression has slow bone remodelling and lower fracture incidence. In contrast, a hypotrophic type is related to reduced BMD, accelerated bone remodelling, high fracture incidence, and reduced thickening of joint cartilage [14]. 

Some determinants, such as genetic, hormonal, and inflammatory factors may account for this correlation. OA and OP have a large number (more than 50%) of genetic components and genes, such as the gene for collagen type 1α1 (COL1A1), the VDR gene, and the estrogen receptor (ER) gene, which take part in the pathogenesis of both diseases [44,45,46]. An interesting consideration is whether people with the genetic predisposition for increased bone density are also predisposed to the rapid onset of OA. The WNT signalling pathway regulates both osteoblasts and osteoclasts, which points to the fact that the same genetic impact on BMD is also present in OA, with separate effects for cartilage. This leads to the conclusion that, in addition to the fact that greater bone mass is conditioned by increased osteoblast activity with increased bone formation, it also affects the development of OA through a greater possibility of osteophyte formation. The latest research focused on genetic factors common to OA and OP points to the importance of the WNT signalling pathway with different genetic variants (LRP5, WNT16, and SOST) and for cathepsin K that have a different role in the homeostasis of cartilage and subchondral bone. Finally, analytical studies of Mendelian randomisation are directed at determining cause and effect relationships between OA and BMD through studies of the most commonly used genomic variants associated with BMD. Systemic factors that affect the relationship between OA and OP are markers of bone metabolism: leptin, osteoprotegerin, and estrogen stimulation and depletion [47,48]. 

Hormonal factors, such as estrogen and leptin, play an important role in bone creation and cartilage homeostasis. Estrogen deficiency can disrupt the differentiation and activation of osteoblasts and osteoclasts, which is very important for the progression of OP [48]. Leptin, which has an anabolic effect on chondrocytes, has a negative effect on bone density in patients with knee OA [47]. The sympathetic nervous system is also involved in the control of bone activity. Sympathetic nerves release vasoactive intestinal peptide and neuropeptide I, which affect osteoblast and osteoclast activity [49]. 

Finally, inflammatory factors, such as interleukin-6 (IL-6) and highly sensitive C-reactive protein [50,51], lead to OA progression and bone loss. Future research should define the relative significance and effects of these factors on OA and osteoporosis [52]. 

Considering the importance of osteoporosis and osteoarthrosis, which are epidemic diseases in the world and our country, the Ministry of Health of the Republic of Serbia launched an amendment to the National Guide for Osteoporosis (2004) intending to standardise diagnostic and therapeutic procedures. For years, we have been working on preventing osteoporosis and osteoarthrosis by conducting preventive examinations at the primary healthcare level, as well as improving treatment and rehabilitation of these patients [53,54]. 

The main limitation of this study was the relatively small cohort of patients, which requires an investigation of additional cases to reinforce the present conclusions. The second limitation pertains to the fact that only females were included. Additional studies should be performed on male patients to support the present conclusions. Future longitudinal studies are needed to confirm these results. 

## 5. Conclusions

Our research has shown that older postmenopausal women with radiological OA of the hip and knee have lower bone mineral density than healthy individuals without OA, which imposes the need for bone mass monitoring in these individuals. A negative correlation was found between age and bone mineral density. These findings point to the need to prevent OA by removing the factors that influence the occurrence of the disease, the need for OA treatment, and the necessity to measure the bone mineral density in these patients with regular monitoring to prevent fractures and preserve the quality of life. 

## Figures and Tables

**Figure 1 medicina-58-01207-f001:**
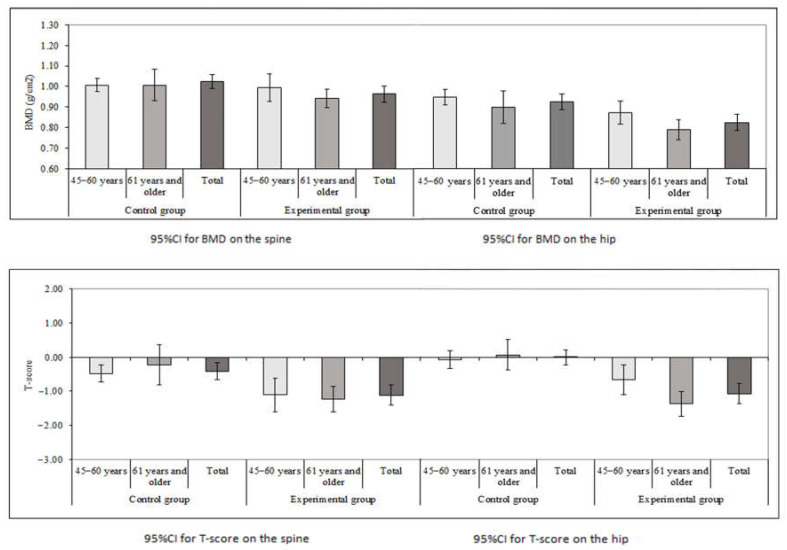
BMD and T-score values on the spine and hip in experimental OA and controls.

**Figure 2 medicina-58-01207-f002:**
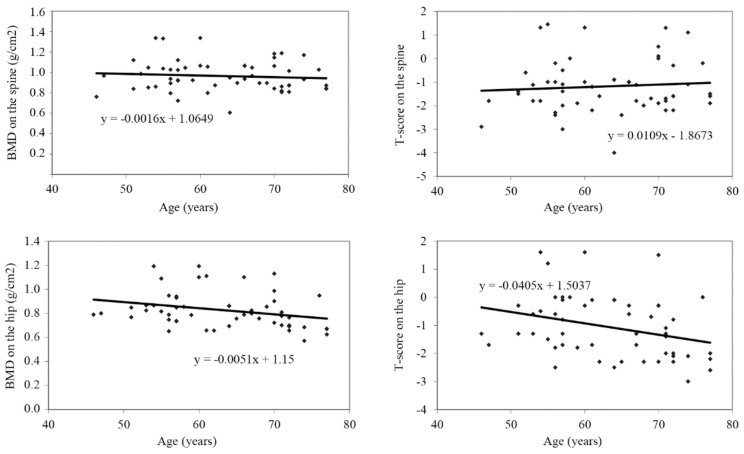
Correlation between age and BMD (g/cm2) and T-score values of the spine and hip.

**Figure 3 medicina-58-01207-f003:**
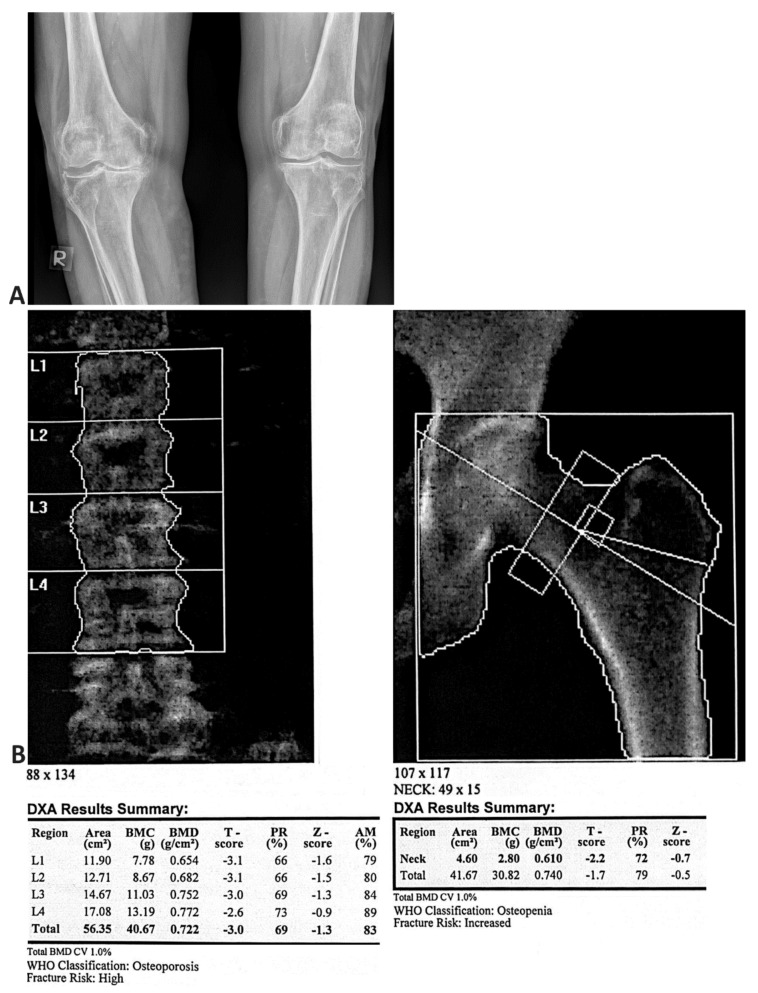
(**A**) posteroanterior radiographic view of bilateral knees demonstrating advanced osteoarthritis. (**B**) the bone density scan (DXA) of a 63-year-old postmenopausal woman with advanced knee osteoarthritis (OA) showing osteoporosis (BMD) of the spine and osteopenia of the hip.

**Table 1 medicina-58-01207-t001:** Baseline characteristics and radiological median grades of OA severity in the experimental and control group.

Group			
Parameters	Control Group (Mean ± SD)	Experimental Group (Mean ± SD)	Comparison of the Groups *p*-Value
Age	60.300 ± 8.125	63.107 ± 8.300	0.148
Height (cm)	166.218 ± 7.212	164.215 ± 6.280	0.307
Weight (kg)	75.380 ± 6.580	73.896 ± 7.498	0.287
BMI (kg/m^2^)	27.767 ± 3.167	27.258 ± 3.428	0.455
Smoking status	(*n* (%))	(*n* (%))	
	10 (25.000%)	15 (26.786%)	0.844 *
Median grades of OA severity (K&L)	Median (Min-Max)	Median (Min-Max)	
Anteroposterior radiographs of the knee	1 (0–1)	2 (2–4)	<0.001
Anteroposterior radiographs of the pelvis	1 (0–1)	2 (2–4)	<0.001

*p*: Student’s *t*-test or Man-Whitney test (depending on data distribution normality for continuous variables); * *p*: chi- 227 square test, (*p* < 0.05, bolded if significant); BMI: body mass index; SD: standard deviation.

**Table 2 medicina-58-01207-t002:** BMD (g/cm^2^) and T-score values of the spine and hip in experimental OA and controls.

Group			
Characteristics	Control Group (Mean ± SD)	Experimental Group (Mean ± SD)	Comparison of the Groups *p*-Value
BMD of the spine			
45–60 years of age	1.007 ± 0.088	0.994 ± 0.169	0.416
61 years of age and older	1.068 ± 0.131	0.942 ± 0.132	0.014
Total	1.024 ± 0.104	0.964 ± 0.150	0.011
BMD of the hip			
45–60 years of age	0.950 ± 0.106	0.874 ± 0.145	0.063
61 years of age and older	0.899 ± 0.134	0.790 ± 0.144	0.024
Total	0.926 ± 0.120	0.826 ± 0.149	0.003
T-score of the spine			
45–60 years of age	−0.483 ± 0.703	−1.110 ± 1.216	0.002
61 years of age and older	−0.227 ± 1.006	−1.232 ± 1.070	0.007
Total	−0.413 ± 0.792	−1.180 ± 1.126	<0.001
T-score of the hip			
45–60 years of age	−0.055 ± 0.737	−0.652 ± 1.089	0.071
61 years of age and older	0.080 ± 0.758	−1.365 ± 1.045	<0.001
Total	0.010 ± 0.731	−1.061 ± 1.112	<0.001

SD: standard deviation, BMD: bone mineral density, *p*: Student’s *t*-test or Man-Whitney test (depending on data distribution normality for continuous variables) (*p* < 0.05, bolded if significant).

**Table 3 medicina-58-01207-t003:** BMD (g/cm^2^) and T-score values of the spine and hip in relation to the radiological median grades of OA severity (K&L) in the experimental group.

RTG Classification (K&L)			
Characteristics	2 (*n* = 31) (Mean ± SD)	3 and 4 (*n* = 25) (Mean ± SD)	Comparison of the Groups *p*-Value
BMD of the spine	1.024 ± 0.140	0.890 ± 0.129	<0.001
BMD of the hip	0.889 ± 0.142	0.746 ± 0.118	<0.001
T-score of the spine	−0.741 ± 0.964	−1.724 ± 1.089	<0.001
T-score of the hip	−0.527 ± 1.045	−1.729 ± 0.797	<0.001

SD: standard deviation, BMD: Bone Mineral Density, *p*: Student’s *t*-test or Man-Whitney test (depending on data distribution normality for continuous variables) (*p* < 0.05, bolded if significant).

**Table 4 medicina-58-01207-t004:** Values of Spearman’s rank correlation coefficient (*ρ*) and 95% CI between age and values of BMD (g/cm^2^) and T-score of the spine and hip in experimental OA.

Age	Statistics	BMD of the Spine	T-Score of the Spine	BMD of the Hip	T-Score of the Hip
	Coefficient	−0.096	0.046	−0.378	−0.349
Total	(95% CI)	(−0.350, 0.171)	(−0.220, 0.305)	(−0.583, −0.128)	(−0.561, −0.095)
	*p*	0.479	0.738	0.005	0.010

BMD: bone mineral density, CI: confidence interval. (*p* < 0.05, bolded if significant).

## Data Availability

The data presented in this study are available on request from the corresponding author.

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
