# Peer review of "Is Osteoarthritis Always Associated with Low Bone Mineral Density in Elderly Patients?"

_medicina, 2022, doi:10.3390/medicina58091207_

Round 1
Reviewer 1 Report (New Reviewer)
This study determined the correlation between hip and knee radiographic OA and bone mineral density (BMD). The comments are as below:
P1-line27: Spine is a countable noun.
P1-line33: Change ‘was’ to ‘were’.
P2-line66: The structure of the sentence is chaotic.
P2-line71: Change ‘has’ to ‘have’.
P2-line78: Change ‘he’ to ‘the’.
P2-line80: The structure of the sentence is chaotic.
P3-line123: Delete as.
P3-line132: Change ‘has’ to ‘have’.
P6-line233: Remove the first colon.
P14-line299: Contribute to doing.
P15-line355: Change ‘osteoblastic’ to ‘osteoblast’.
Author Response
Dear reviewer,
First of all, I would like to thank you for deciding to review our paper.
Please find attached the file with all the corrections as requested
The corrections are highlighted
The English language has been completely revised by two academic experts who are native English speakers. Changes include syntax and semantic corrections.
Sincerely yours,,
Assistant Professor Bojana Stamenkovic

Reviewer 2 Report (New Reviewer)
1. For me, the topic seems very interesting and the title is well-defined and matches the contents. But I read and saw some problems.
2. Corresponding author: 11 *Rančić, K.N; MD, PhD, epidemiologis; For the corresponding author, you must also put an asterisk (*) above, for example Nataša K. Rančić 2,3*
3. The topic seems very interesting and the title is well-defined and matches the contents, but it would be good for your work to mention more citations to strengthen the rationale of your study. You can look for my articles where I discussed about osteoarthritis and I think you will find some interesting aspects, like:
- Gherghel, R., Iordan, D. A., Mocanu, M. D., Onu, A., & Onu, I. Osteoarthritis is not a disease, but rather an accumulation of predisposing factors. A systematic review. Balneo and PRM Research Journal, 12(3), 218-226
- Onu, I., Matei, D., Sardaru, D. P., Cascaval, D., Onu, A., Gherghel, R., ... & Galaction, A. I. (2022). Rehabilitation of Patients with Moderate Knee Osteoarthritis Using Hyaluronic Acid Viscosupplementation and Physiotherapy. Applied Sciences, 12(6), 3165..
4. Age is significantly corelated with 60 the disease, as well as female gender. In the elderly population, OA and fragility fracture 61 due to OP represent a significant health burden. The frequency and mortality of the dis- 62 ease can be reduced by removing the risk factors, primarily by reducing body weight, and 63 preventing the fractures and injuries. 64 Osteoporosis, a metabolic disease characterized by reduced bone mineral density, 65 disorders of the microarchitecture of bones, and increased susceptibility to fractures, also 66 represents a significant health and socioeconomic problem, predominantly for the elderly 67 population. Lower bone density and osteoporosis are common in postmenopausal 68 women due to estrogen deficiency as well as in older men. Worldwide, 200 million people 69 suffer from this disease the most severe consequence of which are fractures. Global disa- 70 bility and mortality from this disease has increased from 207 367 and 8 588 936 cases in 71 1990. to 437 884 and 16 647 466 in 2019. (111.16% and 93.82%), respectively ------ In introduction, who said this??? You??!?! I don t see any citations and I don t belive cause you said this. I am convinced that these aspects can also be found in other works.
5. Tables 1,2,3,4 are placed incorrectly, they must remain and be placed well on the page, I'm waiting for the next version, because there are a lot of spaces left free.
Author Response
Dear reviewer,
First of all, I would like to thank you for deciding to review our paper.
Please find attached the file with all the corrections as requested.
- I corrected the second item you asked (I have put an asterisk above the correspondent’s name)
- I’ve added references that will give more importance to the work. I went through your papers, which are all well-written, and I was happy to refer to them.
- Having been dealing with osteoarthrosis and osteoporosis for more than 20 years, it was not difficult for me to briefly comment on the risk factors and point out the importance of these two diseases from several aspects. However, I have additionally listed references about global burden of osteoporosis and lifetime risk of fractures that supported my thoughts (6,7,8)
- The tables have been corrected according to your suggestion.
The language has been entirely revised by two academic experts who are native English speakers. Changes include syntax and semantic corrections, and grammar and punctuation corrections.
Thank you once again for all the valuable suggestions and comments.
Sincerely yours,
Assistant Professor
Bojana Stamenković

This manuscript is a resubmission of an earlier submission. The following is a list of the peer review reports and author responses from that submission.
Round 1
Reviewer 1 Report
The article "Is Association of Osteoarthritis and Low Bone Mineral Density 2 at the Elderly an Exclusive Appearance?" reports the study's results investigating the correlation between BMD and OA in postmenopausal females. However, the mechanism of found correlations is not explained nor investigated. Analysis of bone turnover or nor microarchitecture could add great value to the presented results. The author should comment why no additional investigation was done.
Author Response
Very Respected Reviewer,
First of all, thank You so much for the cooperation and desire to review my work, as well as for the expediency and quick review. I tried to make adequate corrections to the work.
Research was based on the correlation of moderate to severe radiologically verified osteoarthritis on two locations (knee and hip) in two groups of women different ages, with BMD values on the spine and hip.
It is an introduction to the further analysis of this problem. Pathogenetic mechanisms that explain the connection between these two entities are complex, involving specific changes not only in the cartilage, but also, more important, in the subchondral bone. These changes are different in different types of OA, in early and advanced stages of the disease. According to that statement, monitoring bone remodeling and bone microarchitecture is of great importance, as You mentioned. Our plan is to continue with the research, analyzing bone remodeling markers in different stages of the disease, assessing the fracture risk by FRAX and TBS and fracture prevalence through prospective follow-up. Further steps require a larger number of patients and additional selection.
Thank You, once again, for the useful suggestion and for stimulation further research
Reviewer 2 Report
After reading the manuscript, I have the following remarks:
1. Title: I would recommend changing the title to something like „Is osteoarthritis always associated with low bone mineral density in elderly patients?”
2. The English language needs to be intensively revised, as there are grammatical mistakes that make the manuscript hard to read.
3. Abstract: Please change the Background section in the abstract for a better understanding.
4. Material and Methods: The study includes 120 patients, but there are 56 + 40 = 96. Please explain.
Overall, I do not consider that the manuscript is clinically relevant and does not add new information to the research field.
Author Response
Very Respected Reviewer,
First of all, I would like to thank to You for the cooperation and desire to review my work, as well as for the expediency and quick review. I tried to make an adequate corrections to the manuscript
According to Your suggestion, the text of the paper has been completely revised with the modification of the English language. The abstract was corrected, the method of investigation was supplemented, the conclusion was changed and the section with research results was corrected.
1 The suggestion to change the title of the paper was accepted, as You proposed. The paper has a new title. Thank You.
- The English language has been completely revised by a competent expert in the field
- The background part of the abstract has been changed. I believe that the essence of the problem under consideration is now clearer
- An explanation follows for the material and method section: 120 women were recruited at the beginning of the research for experimental ( with QA knee and hip) and control group (without OA). Data were completed and results obtained for 96 persons of both groups (56+40). Nine female patients were excluded after the radiologist analyzed the hip and knee radiographs, due to technically inadequate finding and impossibility of grading according to K&L. At 15 female patients, an adequate interpretation of the densitometric findings was not obtained due to deformity and degenerative changes in the thoracic spine and artifacts on the spine or hip imaging.
Reviewer 3 Report
Dear authors, thank you for giving us the opportunity to review this paper.
This paper is well written and targets an interesting controversial subject in the medical literature. It finally showed that Osteoarthritis is associated with osteoporosis in elderly, thinking of a possible relationship between both systemic diseases.
This well designed study resolved the controversy shown previously in the literature explaining the apparent relationship between increased BMD and osteoarthritis due to sclerosis and osteophytes. When BMD is assessed when joint narrowing is taken into consideration, and in the appropriate locations, the relationship between OA and decreased BMD becomes obvious.
I would like to recommend some minor modifications.
Mainly,
1- Changing the title to express more the findings of the study.
2- Reviewing the English language in this article by a native English speaker to adjust the structure into more fluid sentences.
3- Add an example showing an x-ray of a patient in the experimental groups together with its corresponding DEXA BMD measurement.
4- Add the 95% interval of confidence for the correlation coefficient in table 2.
Thank you
Author Response
Very Respected Reviewer,
First of all, I would like to thank You for the cooperation and desire to review my work, as well as for the expediency and quick review. I tried to make adequate corrections to the manuscript
- The suggestion to change the title of the paper was accepted, as you proposed. The paper has a new title: “Is osteoarthritis always associated with low bone mineral density in elderly patients? “ Thank You.
- The English language has been completely revised by a competent expert in the field
- We added an example showing an X ray of a patient with knee OA in the experimental group together with the corresponding DXA BMD measurement. Thanks for the suggestion.
- The 95% interval of confidence for the correlation coefficient was added in table 2.
Round 2
Reviewer 1 Report
Thank you for the revised version of the manuscript. I recommend to accept it
Reviewer 2 Report
Dear authors,
Although the manuscript underwent a significant improvement from the English language point of view, and it is much easier to read, I still do not consider that the research is clinically relevant and does not change the current treatment of both pathologies.